# Soft soled footwear has limited impact on toddler gait

Cylie Williams[1]*, Jessica Kolic[1], Wen Wu[2], Kade Paterson[2]

1 School of Primary and Allied Health Care, Faculty of Medicine, Nursing and Health Science, Monash University, Frankston, Victoria, Australia, 2 Department of Physiotherapy, Centre for Health, Exercise and Sports Medicine, School of Health Sciences, Faculty of Medicine Dentistry & Health Sciences, The University of Melbourne, Melbourne, Victoria, Australia

* Cylie.williams@monash.edu

**Data Availability Statement:** All relevant data are within the manuscript and it's supporting information files.

**Funding:** The institutions employing CW and KP received funding from Bobux Pty Ltd (https://www.bobux.co.nz/) for purchase of gait laboratory time

## Abstract

The development of walking in young toddlers is an important motor milestone. Walking patterns can differ widely amongst toddlers, and are characterised by unique biomechanical strategies. This makes comparisons between newly walking toddler's and older children's walking difficult. Little is currently understood regarding the effects of footwear on the gait in newly walking toddlers. A quasi-experimental pre-post study design was used to assess whether spatiotemporal parameters of gait, and in-shoe foot and lower limb kinematics, differed when walking barefoot and in soft-soled footwear in newly walking toddlers. There were 18 toddlers recruited, with 14 undergoing testing. The GAITRite system collected spatial and temporal data. The Vicon camera system collected kinematic data. The testing conditions included barefoot and footwear. Footwear tested was a commercially available soft soled shoe (Bobux XPLORER). Data was extracted directly from the GAITRite system and analysed. Walking in footwear did not change spatial or temporal data, however there were small but significant decreases in hip adduction/abduction range of motion (mean difference (MD) = 1.79˚, 95% CI = -3.51 to -0.07, p = 0.04), knee flexion (MD = -7.63˚, 95% CI = 2.70 to 12.55, p = 0.01), and knee flexion/extension range of movement (MD = 6.25˚, 95% CI = -10.49 to -2.01, p = 0.01), and an increase in subtalar joint eversion (MD = 2.85˚, 95% CI = 5.29 to -0.41, p = 0.03). Effect sizes were small for hip and ankle range, peak knee extension, and subtalar joint ranges (d<0.49), medium for knee flexion/extension range (d = 0.75) and large for peak knee flexion (d = 0.87). The magnitude of kinematic changes with soft-soled footwear were small thus the clinical importance of these findings is uncertain. Future longitudinal studies are needed to develop recommendations regarding footwear for newly walking toddlers.

## Introduction

The emergence of independent upright walking is one of the six fundamental human developmental motor milestones. The behaviour generally appears between the 8th and 18th month of life and is gradually refined with practice and maturity [1]. There is a transition period

and staffing, provision of footwear for testing, participant honorarium, and purchase of consumables used to conduct this study. The funder had no role in study design, data collected and analysis, decision to publish or preparation of the manuscript.

**Competing interests:** The institutions employing CW and KP received funding from Bobux Pty Ltd (https://www.bobux.co.nz/) for purchase of gait laboratory time and staffing, provision of footwear for testing, participant honorarium, and purchase of consumables used to conduct this study. No authors individually received funding other than wages as per their usual employment. This does not alter our adherence to PLOS ONE policies on sharing data and materials.

between crawling and walking where independent walking is refined. During this time, toddlers spend time perfecting standing, side stepping and may practice walking holding a trolley or a parent's hand. It is not until toddlers are walking without this support, are they considered independent walkers [1]. Once an independent walker, gait patterns in toddlers are highly variable, and typically differ greatly from children as young as 4 or 5 years old [2]. For instance, walking speed does not begin to stabilise until approximately 4 years of age. Toddlers with an immature gait pattern also commonly walk with greater knee flexion and greater ankle flexion during loading [2], and this matures to an adult pattern by 2 years of age. While cadence is not considered mature until around 7 years old [3]. Coordination strategies in the first 6 months of walking also vary markedly compared to older children. These strategies may include unique transitory movements such as twisting with, or without falling to propel the child's body forward [4]. Given these important differences, research on the gait patterns of older children, and factors influencing their walking biomechanics, cannot be extrapolated to 'novice walkers' or toddlers.

It is an exciting time for all parents to purchase a first pair of shoes for their toddler. There are numerous types of footwear commercially available in toddler foot sizes, including boots that cover the ankles with firm soles, sandals with variable sole flexibility and limited upper coverage, or pre-walker styles with covered uppers and flexible soles. Yet no studies provide reassurance for parents as to what is the ideal first footwear when presented with this variety. Or if there are any harms from different footwear types. Footwear is one factor that has potential to influence the gait pattern of toddlers. Humans have historically worn footwear for protection while walking [5]. However contemporary footwear choices are complex and influenced by a range of factors including financial, social and cultural pressures [6]. Interestingly, there is limited understanding about the effects of footwear choices on biomechanical factors that may influence foot development, and the emerging gait patterns of toddlers. In contrast, it is well established that wearing shoes changes the walking patterns of older children [7]. In older children, walking in shoes often results in longer steps [8, 9], a faster walking speed [8–10], increased knee sagittal plane range of motion [11], and reduced first metatarsophalangeal joint and three-dimensional motion of the midfoot [12], when compared to walking barefoot. Understanding whether similar changes occur in early walkers is needed to help inform footwear choices for parents and clinicians.

To date, only two studies have investigated the effects of footwear on the gait pattern of a cohort of toddlers [13, 14]. These two studies, on the same cohort of children, report the immediate impact of footwear, with no longitudinal evaluation. Researchers assessed whether footwear torsional flexibility influenced spatial and temporal parameters of gait and plantar pressures. Footwear of interest was categorised by its flexibility relating to the amount of degrees per newtons were required to twist the footwear to $45^{o}$, and tested with a custom built testing jig. Footwear with the highest torsional flexibility (~$70^{o}$/Nm) [14] resulted in a shorter stance time [13], wider step width [13] and higher peak plantar pressures [14] in compared to the most footwear with the stiffer response to torsional testing (~$30^{o}$/Nm). Unfortunately, only very limited information was provided regarding the shoe make, model and specific characteristics other than torsional flexibility. Therefore, it is difficult to generalise the findings to other commercially-available footwear. Furthermore, the study did not assess kinematics, and thus it is unclear how this type of footwear with different flexibility features, affect lower limb biomechanics and foot function within the shoe.

Recommendations for first footwear, range from footwear having as soft a sole as possible to minimise influence of the shoe on muscle development, through to sturdy features to structurally support the foot and assist balance during immature gait and complex tasks [6, 15, 16]. Yet there are limited rigorous investigations of foot function or gait in these different types of

footwear in young children. Therefore, this research aimed to assess whether spatiotemporal parameters of gait, and in-shoe foot and lower limb kinematics, differ when walking barefoot and in soft soled footwear, in newly walking toddlers. This footwear type was chosen given there is no research evaluating its impact on gait, despite widespread use. We hypothesized that soft-soled footwear would result in a difference in the common gait variables, similar to the differences seen in older children walking in footwear compared to walking without footwear.

## Materials and methods

### Study design and ethics

Study design was a quasi-experimental pre-post design following a pre-defined protocol [17]. This research was approved by The Human Research Ethics Committee of Monash University, Victoria Australia (MUHRC 18076). All toddlers participated with written parental consent.

### Participants and setting

Participant recruitment was through social media platforms (Facebook, Twitter), and university newsletters. Participant advertisement was undertaken through social media of the universities. Parents were invited to contact if their toddler was independently walking without parent or equipment support for less than 16 weeks, toddler's foot size matched a shoe that was a European size 20, met all developmental milestones to date, and their toddler had no adverse health events resulting in early intervention for gait problems. Data were collected at the physiotherapy gait laboratory at the University of Melbourne, Victoria, Australia, with all data collected for each participant in a single data collection session.

### Measures and outcomes

Anthropometric measures were collected in the gait laboratory. This included parent reported age, sex, height, weight and weeks since independent walking (defined as ongoing walking greater than 10 steps without parent hand holding for support). Spatial, temporal and kinematic data were collected with the GAITRite® Electronic Walkway (CIR Systems Inc. Haverton, PA, USA) and a 12-camera Vicon MX system (Oxford Metrics Ltd, UK) as explained below. The GAITRite system consists of a 4.3-meter mat, with an active area of measurement that was 427 cm long and 61cm wide. It has a sampling frequency of 80Hz and the active area has 16,128 sensors to collect footprint data and calculate gait spatiotemporal measures.

Kinematic (120 Hz) data were collected with the Vicon system, consisting of 12 cameras that recorded light reflected from 29 markers. We used 14mm sized markers everywhere except the feet, and 9.5mm sized markers at the feet. Markers were placed at the following locations on both limbs: posterior calcaneus, dorsal aspect of the midfoot, medial aspect of the navicular, lateral aspect of the cuboid, medial and lateral malleoli, medial and lateral knee joint, anterior and posterior superior iliac spines, mid-lateral thigh and mid-lateral lower leg shank, right and left anterior aspect of the shoulder, and at C7 at the back of the neck. The marker placement locations were chosen based on a similar protocol recently published by a research team investigating gait acquisition in young children [18]. The marker placement was also similar to the marker position protocols used with older children (S1 Fig) [19].

Marker placement was performed by an experienced certified paediatric podiatrist with 25 years of practice in assessment and treatment of young children with gait problems and supported with a physiotherapist with 1 year of clinical experience. Markers were used to create a custom foot and lower limb kinematic model in Opensim as outlined below. A custom model

was used given there is no accepted kinematic model for this population, and reliability analysis was not performed due to limitations with kinematic set up and data capture in very young children.

## Footwear

A commercially available soft soled shoe was tested (Bobux XPLORER). This shoe was chosen due to its world-wide availability and likeness to other country specific brands, thus improving the generalisability of our findings. A single shoe weighed 30 grams, had a 3mm consistent rear and forefoot sole thickness. There was minimal resistance to longitudinal and torsional bending (i.e. the shoe can be rolled on itself more than $360^{o}$ and completely twisted for the forefoot sole to faced up while the rearfoot sole faced down). It had a leather upper/outer sole, enclosed heel with elastic supporting fit around the rear of the foot and a dorsal strap with Velcro at the front of the shoe. The footwear was modified to accommodate the reflective markers. Small holes were cut into the posterior aspect of the heel, medially over the navicular region, laterally over the cuboid region and dorsally in the midfoot region. No fixtures (i.e. Straps or elastic) were modified when cutting holes in the footwear (S2 Fig) used during data collection.

## Procedure

On entry to the gait laboratory environment, the toddlers were encouraged to explore the testing environment to familiarise themselves with the setting. This included allowing the toddlers to walk freely around the room, play with the research staff, and play with toys set up along the walkway areas. Footwear sizing was checked to ensure appropriate footwear fit under the ankle, the adjustable strap and length being approximately 1cm from the longest toe. During this time, the toddler's legs and clothing were semi-permanently marked for marker placement to ensure any markers that came loose or were removed by the toddler were replaced in the identical position. We preferred toddlers wear only a nappy for testing, however due to variable laboratory temperatures out of our control, some toddlers wore a singlet or upper body covering along with their nappy. All lower limb markers were on bare legs. The reflective markers were adhered to the skin (or shoulder fabric/nappy) over the semi-permanent mark using double sided tape.

We intended to randomise testing order between barefoot and footwear condition. To encourage the toddlers to walk on the GAITrite mat, a posting box was placed approximately 1 meter from the mat end. Toddlers were encouraged to walk to their parent or the posting box at the end of the mat for a minimum of three passes. Where there was a need to entice them to walk, a ball was rolled, or they carried a small and lightweight token in their preferred hand to "post" in the cardboard box. We did not control walking speed however an attempt was made to match speed between trials during the analyses as outlined below. A minimum number of three full passes over the mat was set by the research team to provide a minimum of required steps for data capture. Each toddler was also encouraged to stand still with all markers exposed to motion cameras to capture a static trial used for calibration.

## Analysis

The primary outcome measure was stride length (cm). This variable was chosen as it has commonly been reported to change with footwear worn by children under the age of six [20]. Secondary gait outcome variables, their description and marker placement are in S1 File and S1 Fig. No hierarchical value was placed on these secondary outcomes.

Spatial and temporal data were extracted directly from the GAITRite software. All foot prints were visualized within the software, and any partial foot prints removed prior to

extraction. Gait measures of interest and their description are provided in S1 File. Measured marker trajectories were cleaned, labelled, and then extracted from the Vicon Nexus software and imported into OpenSim software [21]. We attempted to match within-participant walking speed between trials by excluding data collected during slow walking or running trials. The segment lengths of a generic model (built-in model 'Gait2392-Simbody' of OpenSim software) were scaled to those of the toddlers, using the markers captured during the static trial. Inverse kinematics analysis that minimised the difference between the measured markers and those of the scaled models were used to obtain the kinematics of the toddlers. We chose to use this inverse kinematic approach rather than directly calculating kinematic variables from marker trajectories because it has been shown to reduce soft tissue artefacts and inter-tester variations [22, 23]. This was felt to be particularly important for this study given the greater potential for soft tissue artefacts and frequent marker re-attachment of the toddlers during assessment.

The maximum and minimum joint angles for the right hip, knee, ankle, and subtalar joint during the stance phase were reported for each walking condition. While measures were collected for both limbs, there were no statistically significant differences between left and right limb variables. Only measures for the right foot and leg were used in analysis to satisfy the assumption of data independence [19]. Gait variables were described in frequencies (%), means and standard deviations. Differences between barefoot and footwear conditions were analysed with linear regression analysis clustered by individual participant, therefore no height normalisation was used to account for the minimal variations in leg length or height [24]. Robust variance estimates were used to account for the within-subject nature of the data. The mean difference, 95% confidence intervals, p value and effect size [25] were used to understand any differences between walking barefoot and walking in footwear. To determine whether footwear affected walking variability, differences in standard deviations between footwear conditions were compared using paired t tests. Statistical significance was considered as $p<0.05$. A pooled standard deviation was used for this approach to also account for the within subject nature of this data. Cohen's $d$ effect sizes were considered as a secondary statistical variable and categorised as small ($d<0.50$), medium ($d = 0.50$-$0$-$79$), or large ($d\geq0.8$) [26]. No sample size was calculated due to the novel data collection methodology and age group. All data were analysed with Stata 15 [20].

## Results

There were 18 toddlers recruited, with demographic and spatiotemporal data collected from 14 of these toddlers, and kinematic data collected from 13 of the 14 toddlers S2 File. The one toddler without kinematic data refused the marker system required for gait analysis. The remaining four refused any participation either during, or subsequent to marker placement.

The 14 toddlers who participated had a mean (SD) age of 13.3 (2.7) months, a mean (SD) height of 77.6 (3.5) cm, mean (SD) weight of 11.1 (1.1) kg and there were 7 females (50%). The toddlers were walking for a mean (SD) of 7.2 (2.8) weeks. While randomisation of condition order was pre-planned, the challenges of undertaking gait analysis in this age group resulted in the research team taking a pragmatic approach to randomisation. This resulted in eight toddlers wearing footwear first and the remaining toddlers undertaking testing in barefoot first. The toddler's spatiotemporal gait variables and the corresponding effect sizes of walking in soft soled footwear compared to barefoot are displayed in Table 1. Their kinematic variables are displayed in Table 2.

Step length (cm) was the only significantly different spatiotemporal gait variable between walking barefoot and walking in footwear (mean difference (MD) = -2.85cm, 95%CI = 0.31 to 5.39, p = 0.03). Effect sizes for wearing footwear ranged from 0.01–0.68. Velocity (cm/sec) was

**Table 1. Spatiotemporal variables for barefoot and footwear conditions (right side only).**

| GAITRite output | Barefoot Mean | Barefoot Standard Deviation | Footwear Mean | Footwear Standard Deviation | Walking (Barefoot vs Footwear) MD (95%CI) p value | Effect size (*d*) | Standard Deviation p value |
|---|---|---|---|---|---|---|---|
| Velocity (cm/ seconds) | 72.92 | 23.19 | 72.68 | 26.13 | 0.24(-8.78, 8.30) 0.95 | 0.01 | 0.74 |
| Cadence (steps per minute) | 184.73 | 28.83 | 172.35 | 29.21 | 12.38(-27.75, 3.0) 0.11 | 0.42 | 0.19 |
| Stride time (seconds) | 0.66 | 0.10 | 0.78 | 0.24 | -0.12(-0.02, 0.24) 0.08 | 0.63 | 0.18 |
| Stride length (cm) | 46.89 | 9.25 | 51.19 | 12.30 | -4.30 (0.10, 8.51) 0.05 | 0.39 | 0.33 |
| Step time (seconds) | 0.33 | 0.05 | 0.39 | 0.12 | -0.06 (-0.01, 0.12) 0.08 | 0.68 | 0.15 |
| Step length (cm) | 23.11 | 4.50 | 25.96 | 6.46 | **-2.85 (0.31, 5.39) 0.03** | 0.50 | 0.16 |
| Swing percentage (%) | 42.08 | 4.98 | 42.19 | 5.90 | -0.10(-2.55, 2.75) 0.94 | 0.02 | 0.28 |
| Stance percentage (%) | 57.86 | 4.93 | 58.00 | 5.87 | -0.14(-2.47, 2.74) 0.91 | 0.03 | 0.59 |
| Double support time (seconds) | 0.13 | 0.13 | 0.21 | 0.20 | -0.08(-0.04, 0.20) 0.17 | 0.53 | 0.32 |
| Toe in/Toe out (˚) | 4.90 | 6.38 | 5.77 | 5.77 | -0.87(-1.28, 3.02) 0.40 | 0.16 | 0.83 |
| Steps (count) | 69.5 | 31.83 | 64.5 | 33.75 | 5.00(-1061, 10.61) 0.50 | 0.15 | N/A |

Note: Bold figures indicate a significant difference of p<0.05, MD = Mean difference, cm = centimetres.

least effected by wearing footwear (*d* = 0.01), and stride time (seconds) had the highest effect from wearing footwear (*d* = 0.63).

There were some small but statistically significant differences between walking barefoot and walking in footwear in some kinematic variables. Walking in footwear resulted in a significant decrease in hip adduction/abduction range of motion (degrees) (MD = 1.79°, 95% CI = -3.51 to -0.07, p = 0.04), knee flexion (MD = -7.63°, 95% CI = 2.70 to 12.55, p = 0.01), knee flexion/extension range of movement (MD = 6.25°, 95% CI = -10.49 to -2.01, p = 0.01), and greater subtalar joint eversion (MD = 2.85°, 95% CI = 5.29 to -0.41, p = 0.03) compared to walking barefoot. There was also a statistically significant increase in variability when walking barefoot compared to footwear for knee extension (p = 0.04), knee range of motion (p = 0.01) and ankle extension (p = 0.03).

Effect sizes for the hip and ankle range, peak knee extension, and subtalar joint ranges were small (*d*<0.49). While knee flexion/extension range of motion effect size was medium (*d* = 0.75) and peak knee flexion was large (*d* = 0.87).

## Discussion

This is the first study to compare in-shoe foot and lower limb gait kinematics, and spatiotemporal measures of gait in newly walking toddlers. The results showed limited differences in spatiotemporal parameters of gait between conditions, including key variables such as velocity. However, walking in footwear reduced hip frontal plane and knee sagittal plane range of motion, and increased rearfoot eversion, when compared to barefoot walking. Given that the observed differences were generally small, and that there were minimal differences in variability between conditions, the clinical importance of these findings is uncertain.

Only one previous study has investigated limited spatiotemporal measures (velocity, stance time, step width, step length) of toddler's bare foot walking and walking in four different types of footwear [13]. The study reported minimal velocity and step length increases in soft soled footwear. However, velocity changed depending on hardness of shoe sole. Although the

**Table 2. Kinematic variables for barefoot and footwear conditions (right side only).**

| Vicon | Barefoot Mean | Standard Deviation | Shoes Mean | Standard Deviation | Walking (barefoot vs shoes) MD (95%CI) p value | Effect size (*d*) | Standard Deviation p value |
|---|---|---|---|---|---|---|---|
| **Hip** | | | | | | | |
| **Peak flexion (˚)** | 39.00 | 7.27 | 38.21 | 9.25 | 0.79, (-3.87, 2.29) 0.59 | 0.10 | 0.06 |
| **Peak extension (˚)** | 0.49 | 10.03 | -3.23 | 10.62 | 3.73, (-9.80, 2.34) 0.21 | 0.36 | 0.65 |
| **Flexion/extension ROM (˚)** | 38.51 | 8.59 | 41.46 | 9.68 | -2.96, (-2.54, 8.45) 0.26 | 0.33 | 0.34 |
| **Peak adduction (˚)** | 0.59 | 6.00 | -0.34 | 6.62 | 0.93, (-3.13, 1.28) 0.38 | 0.14 | 0.43 |
| **Peak abduction (˚)** | -15.84 | 5.61 | -15.00 | 6.38 | -0.87, (-1.92, 3.67) 0.51 | 0.13 | 0.30 |
| **Adduction/abduction ROM (˚)** | 16.43 | 4.98 | 14.6 | 4.9 | **1.79, (-3.51, -0.07) 0.04** | 0.36 | 0.91 |
| **Peak IR (˚)** | -1.70 | 9.48 | -2.66 ( | 11.35 | 0.96, (-4.82, 2.90) 0.60 | 0.10 | 0.15 |
| **Peak ER (˚)** | -17.33 | 9.61 | -18.96 | 10.59 | 1.63, (-5.64, 2.38) 0.39 | 0.17 | 0.44 |
| **IR/ER ROM (˚)** | 15.64 | 5.78 | 16.31 | 5.07 | -0.67, (-1.92, 3.26) 0.58 | 0.13 | 0.30 |
| **Knee** | | | | | | | |
| **Peak flexion (˚)** | -47.47 | 9.47 | -39.84 | 8.05 | **-7.63, (2.70, 12.55) 0.01** | 0.87 | 0.20 |
| **Peak extension (˚)** | -14.90 | 6.74 | -13.53 | 5.18 | -1.37, (-1.31, 4.05) 0.29 | 0.23 | **0.04** |
| **Flex/ext ROM (˚)** | 32.57 | 9.67 | 26.31 | 6.99 | **6.25, (-10,49, -2.01) 0.01** | 0.75 | **0.01** |
| **Ankle** | | | | | | | |
| **Peak flexion (˚)** | 21.85 | 7.13 | 22.69 | 6.75 | -0.84, (-2.16, 3.83) 0.55 | 0.13 | 0.67 |
| **Peak extension (˚)** | -5.35 | 7.08 | -5.43 | 5.41 | 0.08, (-4.36, 4.20) 0.97 | 0.02 | **0.03** |
| **Flex/ext ROM (˚)** | 27.20 | 8.42 | 28.12 | 7.28 | -0.92, (-3.30, 5.15) 0.64 | 0.11 | 0.25 |
| **Subtalar** | | | | | | | |
| **Peak inversion (˚)** | 21.19 | 15.45 | 19.57 | 14.98 | 1.62, (-4.55, 1.31) 0.25 | 0.10 | 0.80 |
| **Peak eversion (˚)** | 1.05 | 10.25 | -1.79 | 11.40 | **2.85, (-5.28, -0.41) 0.03** | 0.27 | 0.39 |
| **Inversion/eversion ROM (˚)** | 20.14 | 7.32 | 21.36 | 6.07 | -1.21, (-1.18, 3.61) 0.29 | 0.05 | 0.14 |

Note: Bold figures indicate a significant difference of p<0.05.

MD = Mean difference, SD = Standard deviation, ER = external rotation; IR = internal rotation; ROM = range of motion; Flex = flexion; Ext = extension.

authors described variable stiffness of footwear there was limited additional information to understand the footwear fixtures, heel counter, sole composition and shoe weight. Furthermore this study recruited marginally older toddlers (up to two years old) than those in our study [13]. Given the rapid maturation of gait at this age [27] and the lack of information on the footwear under investigation, it is difficult to compare and contrast the spatiotemporal changes with our study which comprised toddlers who were just learning to walk, or to extrapolate findings to other commercially available footwear.

The footwear worn within this study had minimal impact on any spatiotemporal gait parameters. This means, when wearing soft soled light weight shoes with fixtures, toddlers walk similarly to when they were not wearing any footwear. Small kinematic changes were identified when toddlers were walking in soft soled shoes, mainly at the knee. Kinematic changes while toddlers walked in footwear, though small, may be intrinsically linked. For instance, if a "bottom up" theoretical approach is taken, the small, but significant increase in eversion at the subtalar joint while wearing footwear compared to walking bare foot may have been because the toddler unintentionally placed their foot harder on the ground surface to gain sensory input, and orienting themselves to foot placement requirements for propulsion. In turn, this could be linked to reduced knee flexion/extension range of motion and subsequently, reduced abduction/adduction of the hip. In contrast, if we take a "top down"

theoretical approach, footwear may have resulted in the toddlers feeling more secure in not stepping on a surface that may hurt them in any way, therefore enabling more postural stability, reducing their base of gait, resulting in changing in knee flexion and increasing subtalar joint eversion as a result of leg position differences.

There were some limitations that should be considered when interpreting the results of our study. Time spent walking is known to change a number of gait parameters, particularly those including step length, foot clearance and foot placement [28]. Although we were unable to control for this, we attempted to minimise the effects of time spent walking by clustering data by participants during analysis. Furthermore, familiarity (or not) of wearing footwear, and the potential influence of dual-tasking when toddlers carried a small token, may have inadvertently introduced artefacts into gait. Future study designs should consider standardising testing protocols for arm use, clustering participants time spent walking or narrowing the time spent walking timeframe, and possibly harmonising a habituation period of study footwear prior to testing. We were able to collect full amounts of data from the majority of toddlers. This success makes it potentially feasible to collect data on larger numbers of toddlers with the right research personnel and child friendly environment. While training staff and changing the laboratory are easy factors to accommodate in planning gait testing with toddlers, careful planning is required ensuring increased testing time, which factors into increase the cost. This is potentially why this research has not been undertaken before.

Understanding the impact of different types of footwear on early walking is important in order to understand when footwear may play a great role in development or when there is a need for foot and gait support in functional deficit. This is first step to understanding the potential impact of footwear, however the results cannot be extrapolated into footwear recommendations. Marker placement is challenging with this population as we noted markers being removed by the toddlers during testing and while changing footwear. While we attempted to minimise this using semi-permanent marks on the skin to ensure accurate replacement, this may have introduced error. Future research into markerless gait analysis will provide great advantages to measure kinematic movement in this age group. At present, the systems using this technology are focused on perfecting measurement of biomechanical movement in adults and older children, and are limited in how they can evaluate movement in toddlers, or in shoe foot changes.

## Conclusion

Soft soled footwear had minimal effect on joint kinematics and spatiotemporal measures of toddler's gait, compared to walking barefoot. Toddlers walked with a stiffer knee in footwear compared to barefoot, but this could have been as a result of their unfamiliarity with the footwear. Given the relatively small magnitude, the clinical importance of these findings is uncertain. Limited footwear recommendations should arise from this exploratory research though, as toddlers may have different requirements for footwear that are complex and relative to ground surface, task, foot and leg biomechanics, child health and temperature of the environment.

## Supporting information

**S1 Fig. Marker placement.**
(TIF)

**S2 Fig. Footwear modified and unmodified for marker placement.**
(TIF)

**S1 File. Gait variables and their description.**
(DOCX)

**S2 File.**
(XLSX)

## Author Contributions

**Conceptualization:** Cylie Williams, Kade Paterson.

**Data curation:** Cylie Williams, Jessica Kolic.

**Formal analysis:** Jessica Kolic, Wen Wu, Kade Paterson.

**Funding acquisition:** Cylie Williams, Kade Paterson.

**Investigation:** Cylie Williams, Jessica Kolic, Wen Wu, Kade Paterson.

**Methodology:** Cylie Williams, Jessica Kolic, Wen Wu, Kade Paterson.

**Project administration:** Jessica Kolic.

**Resources:** Wen Wu, Kade Paterson.

**Software:** Cylie Williams, Wen Wu, Kade Paterson.

**Supervision:** Cylie Williams, Kade Paterson.

**Validation:** Cylie Williams, Kade Paterson.

**Writing – original draft:** Cylie Williams, Kade Paterson.

**Writing – review & editing:** Cylie Williams, Jessica Kolic, Wen Wu, Kade Paterson.

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
