## [Decision Letter · Decision Letter 0]

3 Jan 2021

PONE-D-20-25753

Soft soled footwear has limited impact on toddler gait

PLOS ONE

Dear Dr. Williams,

Thank you for submitting your manuscript to PLOS ONE. After careful consideration, we feel that it has merit but does not fully meet PLOS ONE’s publication criteria as it currently stands. Therefore, we invite you to submit a revised version of the manuscript that addresses the points raised during the review process.

We look forward to receiving your revised manuscript.

Kind regards,

Riccardo Di Giminiani

Academic Editor

PLOS ONE

Journal Requirements:

3.Thank you for stating the following in the Financial Disclosure section:

"CW and KP received funding from Bobux Pty Ltd (https://www.bobux.co.nz/). The funder had no role in study design, data collected and analysis, decision to publish or preparation of the manuscript. "

We note that you received funding from a commercial source: Bobux Pty Ltd

Reviewers' comments:

Reviewer's Responses to Questions

**Comments to the Author**

1. Is the manuscript technically sound, and do the data support the conclusions?

Reviewer #1: Partly

Reviewer #2: Yes

2. Has the statistical analysis been performed appropriately and rigorously? 

Reviewer #1: No

Reviewer #2: Yes

3. Have the authors made all data underlying the findings in their manuscript fully available?

Reviewer #1: Yes

Reviewer #2: Yes

4. Is the manuscript presented in an intelligible fashion and written in standard English?

Reviewer #1: Yes

Reviewer #2: Yes

5. Review Comments to the Author

Reviewer #1: 1. Introduction:

a. For children, the selection of footwear ranges from very soft to very stiff. But what about the options for infants? Is there very stiff footwear for infants on the market? Is that popular and beneficial to infants?

b. The authors cited two studies from a same toddler study which compared the softest footwear and the stiffest footwear. How were these two footwears different from a mechanical test?

c. The author agreed that different shoe soles can lead to different walking patterns in children. However, there is no rationale on why soft shoe sholes were selected for this study.

d. Review of gait development in newly walking infants is not complete.

e. There is no hypothesis to test in this study.

2. Methods:

a. Participants were toddlers with walking experience less than 16 weeks. What was the definition of walking onset? Over the first few months after walking onset, gait pattern changes drastically in newly walking toddlers. So, it may not be appropriate to consider all the subjects in a homogeneous group for data analysis.

b. Marker placement: Have this marker placement been used by any previous studies? Cite any references if possible.

c. Apparently, there are many shoe brands on the market. Provide a rationale on the selection of these commercially available shoes.

d. All the variables in Table 1 are common gait variables. So, this table should be included as a supplemental table in the Appendix.

e. Did you run a preliminary analysis between the left and right sides and determine that there was no statistical difference between the two sides in the dependent variables?

f. Some references are needed to support the rationale on not normalizing the data by height or leg length.

g. What was the statistical model? What was the significance level?

3. Results:

a. Walking experience 7.2 (SD 2.8) weeks can be a big confounding factor in data interpretation.

b. Effect size should be a secondary statistical variable.

4. Discussion:

a. Velocity is an important gait variable and indicates walking ability. It seems that footwear does not change velocity; rather, it modifies some gait variables.

Reviewer #2: The authors gather biomechanics data from (adorable!) toddlers with and without shoes, and look for differences in these data. As this area has had little research, the study is exploratory and did not, I believe, have a priori hypotheses. The no doubt quite difficult to carry out study is well designed and the topic is novel.

The work is publishable with one potentially major issue and the rest minor.

The only major issue is whether a Bonferroni or other multiple comparisons correction should have been applied to the statistics. A lot of tests are made; should their p-values not be corrected? As this is not my primary area, I do not know the answer with certainly - would like to hear the authors' view. Even if this removed the significance of, e.g., the knee flexion, the results are still interesting and should be published, with commentary in dicussion as it is speculating about reasons for the observed changes.

Minor things to consider:

p8: are toddler bodies isometrically scaled version of adult bodies? i think they may be slightly different in arm lengths. does this mean the applications of the generic model could affect the results? Presumably if the same model used for all subjects and both conditions, there is less likely a bias.

p9: does getting the kinematics from fitted models suppress any important varation? I understand individual models is definitely not possible!

p9 typo - maker -> marker

p15: grammar: "given difference" ->"Given that the observed differences..."

6. PLOS authors have the option to publish the peer review history of their article (what does this mean?). If published, this will include your full peer review and any attached files.

Reviewer #1: No

Reviewer #2: No

---

## [Author Response · Author response to Decision Letter 0]

6 Jan 2021

Reviewer #1: 1. Introduction:

a. For children, the selection of footwear ranges from very soft to very stiff. But what about the options for infants? Is there very stiff footwear for infants on the market? Is that popular and beneficial to infants?

Response: There are variable footwear types commercially available for young children. We have detailed within the methods that the size of commercially available footwear tested was a EU Sz 20. At the time of this response, we are able to find within common online footwear stores a variety of footwear in this size ranging from sneakers, boots, joggers, gumboots, sandals and similar styles shoes to those tested within this research. We have included a statement at Page 3, Lns 78-80, highlighting the variety available to parents: 

There are numerous types of footwear commercially available in toddler foot sizes, including boots that cover the ankles with firm soles, sandals with variable sole flexibility and limited upper coverage, or pre-walkers styles with covered uppers and flexible soles.

b. The authors cited two studies from a same toddler study which compared the softest footwear and the stiffest footwear. How were these two footwears different from a mechanical test?

Response: 

Additional information is provided about the footwear within these two studies on Pages 4-5, Lns 102-114. We have added: 

Footwear of interest was categorised by its flexibility relating to the amount of degrees per newtons were required to twist the footwear to 45o, and tested with a custom built testing jig. Footwear with the highest torsional flexibility (~70o/Nm) (14) resulted in a shorter stance time (13), wider step width (13) and higher peak plantar pressures (14) in compared to the most footwear with the stiffer response to torsional testing (~30o/Nm).

c. The author agreed that different shoe soles can lead to different walking patterns in children. However, there is no rationale on why soft shoe sholes were selected for this study.

Response: 

We have included the statement on Page 5 Ln 128: 

This footwear type was chosen given there is no research evaluating its impact on gait, despite widespread use. 

d. Review of gait development in newly walking infants is not complete.

Response: 

Although we are unsure of which aspects of gait development the reviewer is alluding to, we have made a number of amendments to our first paragraph to provide a greater description of the development of gait in newly walking infants. We hope that the reviewer finds this satisfactory.

We have provided details on age of gait acquisition, velocity, cadence, arm swing through transitory movements and coordination patterns. We have also added discussion about the joint angles with the following statements on Page 3 Lns 61-65 and Lns 68-70: 

There is a transition period between crawling and walking where independent walking is refined. During this time, toddler’s spend time perfecting standing, side stepping and may practice walking holding a trolley or a parent’s hand. It is not until toddlers are walking without this support, are they considered independent walkers (1). 

AND

Toddlers with an immature gait pattern also commonly walk with greater knee flexion and greater ankle flexion during loading (2), and this matures to an adult pattern by 2 years of age. 

e. There is no hypothesis to test in this study.

Response:

We have included the statement at Page5, Lns 129-131: 

We hypothesized that soft-soled footwear would result in a difference in the common gait variables, similar to the differences seen in older children demonstrating confident walking in footwear compared to walking without footwear.

2. Methods:

a. Participants were toddlers with walking experience less than 16 weeks. What was the definition of walking onset? Over the first few months after walking onset, gait pattern changes drastically in newly walking toddlers. So, it may not be appropriate to consider all the subjects in a homogeneous group for data analysis.

Response:

We have clarified independent walking in the introduction and have included this in the methods section on Page 6, Line 143. We defined walking with the parent’s during recruitment as total walking independence without parental or equipment support on Page 6, Ln 144.

b. Marker placement: Have this marker placement been used by any previous studies? Cite any references if possible.

Response: As we have outlined in our methods (Page 7, Lines 166-169), there is no accepted kinematic model for this population. To our knowledge, only one other research team has used a similar marker model in this age group, however they have not published results as yet. To acknowledge this, we have cited their research protocol and provided the following statement: 

The marker placement locations were chosen based on a similar protocol recently published by a research team investigating gait acquisition in young children (17). The marker placement was also similar to the marker position protocols used with older children (18).

c. Apparently, there are many shoe brands on the market. Provide a rationale on the selection of these commercially available shoes.

Response:

We have provided this based on your suggestion both in the introduction as previously requested but also an additional statement (Page 7, Lines 180-182) within the footwear section. 

This shoe was chosen due to its world-wide availability and likeness to other country specific brands, thus improving the generalisability of our findings.

d. All the variables in Table 1 are common gait variables. So, this table should be included as a supplemental table in the Appendix.

Response: We have moved this data to supplementary files based on your suggestion and referenced this at Page 9, Line 228. 

e. Did you run a preliminary analysis between the left and right sides and determine that there was no statistical difference between the two sides in the dependent variables?

Response: We did as per our protocol and have added this statement at Page 10, Line 225-226. 

While measures were collected for both limbs, there were no statistically significant differences between left and right limb variables.

f. Some references are needed to support the rationale on not normalizing the data by height or leg length.

Response: 

We have analysed the data in a way that adjusting for variables like the minimal variation in height is unnecessary as we have only ‘tested’ the child against themselves. We have described this within the analysis through the statement regarding clustering by participant within the linear regression model, and have now provided a reference for this type of analysis, and its assumptions (Page 10, Line 251). This method of analysis has been used in other gait papers for within participant child related data with a single time-point/short term intervention including: 

- Cranage S, et al. A comparison of young children’s spatiotemporal measures of walking and running in three common types of footwear compared to and bare feet. Gait Posture. 2020; Volume 81, Pages 218-224

- Williams C.M, et al, 2016, Whole body vibration results in short-term improvement in the gait of children with idiopathic toe walking, J Child Neuro, 31 (9), 1143-1149

- Michalitsis J, et al, 2019 Full length foot orthoses have an immediate treatment effect and modify gait of children with idiopathic toe walking, Gait Post 68, 227-231

g. What was the statistical model? What was the significance level?

Response: 

This has been provided within the manuscript (Page 10, Line 254). We have also added the following sentence: 

Statistical significance was considered as p<0.05

3. Results:

a. Walking experience 7.2 (SD 2.8) weeks can be a big confounding factor in data interpretation.

Response: 

We agree and have discussed this in great detail as a limitation within the discussion section of the paper (Page 16, Lines 342-345). We have also provided additional suggestions for future research with this age group in how to minimise the variability with toddler gait. As limited research has been done in this age group with this type of gait analysis, we were unsure about the feasibility of gait analysis with this age group and therefore recruited a wider age group. We have added recommendations for future research to cluster participants by time spent, or narrow the time spent walking this now that we know this research is feasible (Page 16, Line 349)

b. Effect size should be a secondary statistical variable.

Response: 

We agree, and have further highlighted this within the data analysis section. No change has been made in the results. We have included a statement (Page 10, Ln 258) within the data analysis section reading: 

Cohen’s d effect sizes were considered as a secondary statistical variable and categorised as small (d<0.50), medium (d=0.50-0-79), or large (d>0.8) (25).

4. Discussion:

a. Velocity is an important gait variable and indicates walking ability. It seems that footwear does not change velocity; rather, it modifies some gait variables.

Response: 

We agree that velocity is an important indicator of walking ability, it has been observed as changing with footwear in older children and adults. Thus, the finding of no difference in early walkers is indeed interesting. We have highlighted this in our opening paragraph in the discussion (Page14, Line 305) as follows:

“The results showed limited differences in spatiotemporal parameters of gait between conditions, including key variables such as velocity.”

Reviewer #2: The authors gather biomechanics data from (adorable!) toddlers with and without shoes, and look for differences in these data. As this area has had little research, the study is exploratory and did not, I believe, have a priori hypotheses. The no doubt quite difficult to carry out study is well designed and the topic is novel.

Response: 

We thank the reviewer for their encouraging kind words. We are in total agreement that this project recruited adorable toddlers. It was indeed a very challenging protocol and we have now included the protocol reference approved by our ethics committee (Page 5, Line 134) and described where it deviated (Page 11, Line 282), however one we refined and are confident could be replicated based on how we have described it and it’s modifications with this paper. Please note, in response to feedback from Reviewer #1 we have now included our original broad hypothesis.

The work is publishable with one potentially major issue and the rest minor.

The only major issue is whether a Bonferroni or other multiple comparisons correction should have been applied to the statistics. A lot of tests are made; should their p-values not be corrected? As this is not my primary area, I do not know the answer with certainly - would like to hear the authors' view. Even if this removed the significance of, e.g., the knee flexion, the results are still interesting and should be published, with commentary in discussion as it is speculating about reasons for the observed changes.

Response: 

Thank you for the opportunity to consider this question and rebut why we have not used this correction. There are times that the Bonferroni correction could be considered, and used in gait analysis research. However, it would not be appropriate to use it with this study for a number of reasons based on the design, hypothesis (a knowledge of impact of footwear on older children’s gait) and how we handled the data during analysis. 

Firstly, the Bonferroni or similar multiple comparison tests are used to reduce the chance of Type I errors during ANOVA or MANOVA. We analysed this data with linear regression models as discussed in reference Reviewer 1’s question on normalisation. This alternative method of analysis allowed us to minimise the introduction of Type I errors during the analysis itself with the use of a robust variance estimates. The use of the Bonferroni's correction in this case would most likely introduce even more Type II errors (Rothman, 1990). 

Secondly, this correction method should only be considered based on the hypothesis (and it’s outcomes) and study design. In reference to our study, we had a pre-planned aim and hypothesis, in particular, that our hypothesis being we would have variable differences in gait outcomes between wearing footwear and not wearing footwear. Bonerroni corrections should be considered when the null hypothesis is that all variables would be, or all not be, different (Perneger, 1998)

References: 

Rothman KJ. No adjustments are needed for multiple comparisons. Epidemiology. 1990;1:43–46. 

Perneger TV. What's wrong with Bonferroni's adjustment. BMJ 1998; 316: 1236–1238.

Minor things to consider:

p8: are toddler bodies isometrically scaled version of adult bodies? i think they may be slightly different in arm lengths. does this mean the applications of the generic model could affect the results? Presumably if the same model used for all subjects and both conditions, there is less likely a bias.

Response: 

This assumption of arm length difference is correct for toddlers compared to older children and adults. However, please note that we have not used any variable or marker for arm length therefore this is unlikely to affect our results. 

Torso/leg length ratio is also a scaling difference between toddlers, older children and adults, but as the reviewer rightly points out, we are not comparing data between different age groups but to similar children of similar size and torso/length ratio. We have discussed this in more detail on page 10, Line 234. 

The segment lengths within the generic model was therefore adjusted and customised model to account these differences, as no other models fitting this size have been developed. We would also highlight that we tested this model to minimise the difference between the scaling and measured markers. We have referenced the rational for doing this as part of standard practice where participant anthropometrics do not match the model (Page 10, Line 239). We have not amended the manuscript in response to this and hope this allays the reviewer’s concern. 

p9: does getting the kinematics from fitted models suppress any important varation? I understand individual models is definitely not possible!

Response: The kinematic model was participant-specific as we scaled the segments length of a generic model to those of each participant using the markers captured during the static trial. We reworded the following statement on Page 11 to better clarify this.

The segment lengths of a generic model (built-in model ‘Gait2392-Simbody’ of OpenSim software) were scaled to those of the toddlers using the markers captured during the static trial.

Furthermore, we employed a within-subjects design and analysis clustered by individual participants. These methodological features ensured we captured and retained as much individual variation within our analyses as practicable.

p9 typo - maker -> marker

p15: grammar: "given difference" ->"Given that the observed differences..."

Response: 

These type/grammar errors have been amended.

---

## [Decision Letter · Decision Letter 1]

5 Mar 2021

PONE-D-20-25753R1

Soft soled footwear has limited impact on toddler gait

PLOS ONE

Dear Dr. Williams,

Thank you for submitting your manuscript to PLOS ONE. After careful consideration, we feel that it has merit but does not fully meet PLOS ONE’s publication criteria as it currently stands. Therefore, we invite you to submit a revised version of the manuscript that addresses the points raised during the review process.

We look forward to receiving your revised manuscript.

Kind regards,

Riccardo Di Giminiani

Academic Editor

PLOS ONE

Journal Requirements:

Reviewers' comments:

Reviewer's Responses to Questions

**Comments to the Author**

1. If the authors have adequately addressed your comments raised in a previous round of review and you feel that this manuscript is now acceptable for publication, you may indicate that here to bypass the “Comments to the Author” section, enter your conflict of interest statement in the “Confidential to Editor” section, and submit your "Accept" recommendation.

Reviewer #2: All comments have been addressed

Reviewer #3: (No Response)

2. Is the manuscript technically sound, and do the data support the conclusions?

Reviewer #2: Yes

Reviewer #3: Yes

3. Has the statistical analysis been performed appropriately and rigorously? 

Reviewer #2: Yes

Reviewer #3: Yes

4. Have the authors made all data underlying the findings in their manuscript fully available?

Reviewer #2: (No Response)

Reviewer #3: Yes

5. Is the manuscript presented in an intelligible fashion and written in standard English?

Reviewer #2: (No Response)

Reviewer #3: Yes

6. Review Comments to the Author

Reviewer #2: (No Response)

Reviewer #3: Abstract

- The authors mention kinetic data in the abstract but don’t provide information on how these data were acquired (i.e., they state that they used GAITRite to collect spatiotemporal data and Vicon to collect kinematic data, but not how they acquired kinetic data).

Introduction

- Line 48: remove the apostrophe in toddlers (i.e., it should read toddlers not toddler’s).

- Line 107: What do the authors mean by “confident” walking? This is unclear.

Method

- Did the authors ask parents whether and how often their toddlers wore their shoes at home? I wonder if some toddlers may have more experience walking in shoes than others.

- Again, in the abstract, there was mention made of kinetic data, but this is not in the method section.

- Please clarify the clothing on which markers were placed (line 176). Were all toddlers wearing a onsie? I ask because markers placed directly onto clothing can move easily and possibly not reflect true body movement.

- Please explain why stride length was used instead of step length. Literature shows that newly walking toddlers demonstrate variability from step to step suggesting that step length is a better measure to use.

Results

- On line 245, information is provided about weeks of walking experience. In the method section, the authors should provide details about how they acquired these data (e.g., Were parents asked to report this during an interview? If so, what definition was provided to the parents about what walking onset means?).

- Standard deviations for the spatiotemporal parameters appear to be larger for toddlers when they walked wearing shoes. Is that variability significantly different from when toddlers walked barefoot? Variability is also a very important factor to examine when studying newly walking toddlers.

Discussion

- Lines 325 – 339: Although mean differences in spatiotemporal parameters were not different, I still question whether variability differed between barefoot and shod walking (i.e., differences in standard deviations).

7. PLOS authors have the option to publish the peer review history of their article (what does this mean?). If published, this will include your full peer review and any attached files.

Reviewer #2: No

Reviewer #3: No

---

## [Author Response · Author response to Decision Letter 1]

7 Apr 2021

7th April 2021

Riccardo Di Giminiani

Academic Editor, PLOS One

Dear A/Prof Di Giminiani, 

Re: Manuscript ID: PONE-D-20-25753

We thank you and the reviewers for their comments and for the opportunity to revise our manuscript. What follows is a point by point response to each comment for Reviewer 3. We also thank Reviewer 2 for their positive comments. 

Reviewer #3: 

1. The authors mention kinetic data in the abstract but don’t provide information on how these data were acquired (i.e., they state that they used GAITRite to collect spatiotemporal data and Vicon to collect kinematic data, but not how they acquired kinetic data).

Response

Thank you for finding this within the abstract. We have deleted this comment as it was added in error, and therefore the reason it was not presented within the manuscript. 

- Line 48: remove the apostrophe in toddlers (i.e., it should read toddlers not toddler’s).

Response

Amended

- Line 107: What do the authors mean by “confident” walking? This is unclear.

Response

On reflection, this term could create confusion. We have removed it as it relates more to older children and their walking

- Did the authors ask parents whether and how often their toddlers wore their shoes at home? I wonder if some toddlers may have more experience walking in shoes than others.

Response

We did not consider this during the design phase of our research. We would highlight that we did consider home shoe use during the interpretation of our results, and we alluded that familiarity with footwear may have introduced artifacts or variability (please see Lines 363-364). We have also encouraged researchers undertaking research with todders in the future to consider habituation with footwear to minimise the possibility of home shoe use influencing study outcomes (Lines 365-368). 

- Again, in the abstract, there was mention made of kinetic data, but this is not in the method section.

Response

As previously mentioned, we did not collect this data. Its inclusion in the abstract was an error and has now been removed. 

- Please clarify the clothing on which markers were placed (line 176). Were all toddlers wearing a onsie? I ask because markers placed directly onto clothing can move easily and possibly not reflect true body movement.

Response

Thank you for the opportunity to clarify this. There were no markers on clothing on the lower limb, only the shoulders (for some children who did not want their singlet removed) and nappy area. We have added the following at Lines 184-188 to better explain this:

We preferred toddlers wear only a nappy for testing, however due to variable laboratory temperatures our of our control, some toddlers wore a singlet or upper body covering along with their nappy. All lower limb markers were on bare legs. The reflective markers were adhered to the skin (or shoulder fabric/nappy) over the semi-permanent mark using double sided tape.

- Please explain why stride length was used instead of step length. Literature shows that newly walking toddlers demonstrate variability from step to step suggesting that step length is a better measure to use.

Response

We used stride length to facilitate comparison with previously published data. On reflection, we now have provided right limb step length and time within Table 1, and amended the results section in lines 288-290: We have also provided additional information to understand any variability in response to the comment below. 

Step length (cm) was the only significantly different spatiotemporal gait variable between walking barefoot and walking in footwear (mean difference (MD) = -2.85cm, 95%CI= 0.31 to 5.39, p=0.03).

- On line 245, information is provided about weeks of walking experience. In the method section, the authors should provide details about how they acquired these data (e.g., Were parents asked to report this during an interview? If so, what definition was provided to the parents about what walking onset means?).

Response

We have now added more detail regarding the instructions we provided parents on how to describe independent walking at Line 131-132:

This included parent reported age, sex, height, weight and weeks since independent walking (defined as ongoing walking greater than 10 steps without parent hand holding for support).

- Standard deviations for the spatiotemporal parameters appear to be larger for toddlers when they walked wearing shoes. Is that variability significantly different from when toddlers walked barefoot? Variability is also a very important factor to examine when studying newly walking toddlers.

Response

Thank you for this suggestion. We have now compared the standard deviations between the shod conditions for all variables and reported the p value of these comparisons in Tables 1 and 2. We have also discussed the findings in lines 290-291 and 301-303 of the results. We have also commented on the variables and their unknown clinical significance within the discussion at lines 326-327. During this time, we also took the opportunity to ensure there was consistency in decimal points between the tables as there were some small differences in rounding from the previous submission. 

- Lines 325 – 339: Although mean differences in spatiotemporal parameters were not different, I still question whether variability differed between barefoot and shod walking (i.e., differences in standard deviations).

Response

As per previous request we have provided these in the table and further information in results. 

Kind regards, 

Cylie Williams, on behalf of the author team.

---

## [Decision Letter · Decision Letter 2]

22 Apr 2021

Soft soled footwear has limited impact on toddler gait

PONE-D-20-25753R2

Dear Dr. Williams,

We’re pleased to inform you that your manuscript has been judged scientifically suitable for publication and will be formally accepted for publication once it meets all outstanding technical requirements.

Kind regards,

Riccardo Di Giminiani

Academic Editor

PLOS ONE

Additional Editor Comments (optional):

Reviewers' comments:

Reviewer's Responses to Questions

**Comments to the Author**

1. If the authors have adequately addressed your comments raised in a previous round of review and you feel that this manuscript is now acceptable for publication, you may indicate that here to bypass the “Comments to the Author” section, enter your conflict of interest statement in the “Confidential to Editor” section, and submit your "Accept" recommendation.

Reviewer #3: All comments have been addressed

2. Is the manuscript technically sound, and do the data support the conclusions?

Reviewer #3: Yes

3. Has the statistical analysis been performed appropriately and rigorously? 

Reviewer #3: Yes

4. Have the authors made all data underlying the findings in their manuscript fully available?

Reviewer #3: Yes

5. Is the manuscript presented in an intelligible fashion and written in standard English?

Reviewer #3: Yes

6. Review Comments to the Author

Reviewer #3: The authors have satisfactorily addressed the reviewers' comments. I think that the paper is ready to be accepted.

7. PLOS authors have the option to publish the peer review history of their article (what does this mean?). If published, this will include your full peer review and any attached files.

Reviewer #3: No

---

## [Editor Report · Acceptance letter]

29 Apr 2021

PONE-D-20-25753R2 

Soft soled footwear has limited impact on toddler gait 

Dear Dr. Williams:

I'm pleased to inform you that your manuscript has been deemed suitable for publication in PLOS ONE. Congratulations! Your manuscript is now with our production department. 

Kind regards, 

on behalf of

Prof. Riccardo Di Giminiani 

Academic Editor

PLOS ONE